

# Candidate genes for shell colour polymorphism in *Cepaea nemoralis*

Jesse Kerkvliet[1], Tjalf de Boer[2,3], Menno Schilthuizen[4] and Ken Kraaijeveld[3,5]

[1] Bio-informatics, University of Applied Sciences Leiden, Leiden, The Netherlands
[2] MicroLife Solutions, Amsterdam, The Netherlands
[3] Department of Ecological Science, Vrije Universiteit Amsterdam, Amsterdam, The Netherlands
[4] Naturalis Biodiversity Center, Leiden, The Netherlands
[5] Leiden Genome Technology Center, Leiden University Medical Center, Leiden, The Netherlands

## ABSTRACT

The characteristic ground colour and banding patterns on shells of the land snail *Cepaea nemoralis* form a classic study system for genetics and adaptation as it varies widely between individuals. We use RNAseq analysis to identify candidate genes underlying this polymorphism. We sequenced cDNA from the foot and the mantle (the shell-producing tissue) of four individuals of two phenotypes and produced a *de novo* transcriptome of 147,397 contigs. Differential expression analysis identified a set of 1,961 transcripts that were upregulated in mantle tissue. Sequence variant analysis resulted in a set of 2,592 transcripts with single nucleotide polymorphisms (SNPs) that differed consistently between the phenotypes. Inspection of the overlap between the differential expression analysis and SNP analysis yielded a set of 197 candidate transcripts, of which 38 were annotated. Four of these transcripts are thought to be involved in production of the shell's nacreous layer. Comparison with morph-associated Restriction-site Associated DNA (RAD)-tags from a published study yielded eight transcripts that were annotated as metallothionein, a protein that is thought to inhibit the production of melanin in melanocytes. These results thus provide an excellent starting point for the elucidation of the genetic regulation of the *Cepaea nemoralis* shell colour polymorphism.

# INTRODUCTION

Since the first studies of selection on the banding patterns and colours on the shell of the land snail *Cepaea nemoralis* over 60 years ago (*Cain & Sheppard, 1950*; *Cain & Sheppard, 1952*; *Cain & Sheppard, 1954*), the polymorphism has become a classic study system for genetics and adaptation (*Jones, Leith & Rawlings, 1977*; *Cook, 1998*; *Silvertown et al., 2011*; *Cameron & Cook, 2012*). For example, shell colour affects habitat-dependent predation risk and thermoregulation (*Lamotte, 1959*; *Clarke, 1962*; *Arnold, 1971*; *Greenwood, 1974*). More recently, the *Cepaea* shell colour polymorphism became the subject of the citizen science project Evolution MegaLab, in which citizen scientists are asked to score the phenotypes of *Cepaea* snails in their surroundings and upload their records to the MegaLab website (*Worthington et al., 2012*). The aim of this project is to show that (possibly human-induced)

Corresponding author
Jesse Kerkvliet, jesse@kerkvliet.info

selection differences can result in differences in allele frequencies on a local as well as a continental scale (*Silvertown et al., 2011*; *Schilthuizen, 2013*).

Despite the prominence of the *Cepaea* study system in scientific discourse as well as public outreach, little is known about the underlying molecular genetic machinery that produces the different phenotypes. The *Cepaea* shell polymorphism consists of nine phenotypic traits, which include shell ground colour and various aspects of the banding pattern (*Richards et al., 2013*). Genes underlying five of these traits are closely linked to form a so-called supergene (*Schwander, Libbrecht & Keller, 2014*), which inherit with very little recombination between the alleles, keeping the alleles together as one large gene. (*Schwander, Libbrecht & Keller, 2014*). *Richards et al. (2013)* identified eleven restriction-site associated DNA (RAD) tags that were linked to the supergene. Three of these tags were within ∼0.6 cM of the supergene. *Mann & Jackson (2014)* characterized the major proteinaceous components of the *C. nemoralis* shell, but were unable to identify proteins associated with shell pigmentation. The aim of our work is to identify candidate genes that may underlie the *Cepaea* shell polymorphism.

To identify candidate genes that play a role in the polymorphism, the RNA of four juvenile *C. nemoralis* individuals (two brown/unbanded and two yellow/banded) was sequenced. For each individual, we sequenced the transcriptome of the mantle (the organ in which the shell is formed) and the snail foot. Since the polymorphism is only visible in the shell, we focus on candidate genes that are upregulated in the mantle. Within this set of mantle-specific genes, we search for sequence variants that differ consistently between different phenotypes. Furthermore, we search for the three supergene-associated RAD tags reported by *Richards et al. (2013)*, as they are found in near proximity of the supergene. Our results provide the first clues to the molecular mechanisms underlying the *Cepaea* polymorphism and will provide a starting point for elucidating the genetic architecture of this classic polymorphism.

## METHODS

### Sample collection, mRNA extraction, sequencing and assembly

Four juvenile *C. nemoralis* with different phenotypes (two with brown shell and no banding and two with a yellow shell with multiple dark bands; Fig. S1) were collected at the Van Veldhuizenbos near Dronten, The Netherlands. Total RNA was extracted separately from the mantle and the foot for each of the four *C. nemoralis* individuals using the NucleoSpin RNA kit (Macherey-Nagel, Düren, Germany), following the manufacturer's protocol. The remains of the specimens have been deposited in the alcohol collection of Naturalis Biodiversity Center under reference numbers RMNH.5004228-5004231. The total RNA was subjected to polyA selection, converted to cDNA and used to generate sequencing libraries as described in *Salazar-Jaramillo et al. (2017)*. The libraries were paired-end 2 × 100 bp sequenced on an Illumina HiSeq 2000 at the Leiden Genome Technology Center.

### *De novo* assembly, protein prediction and annotation

Sequence reads of all eight samples were combined to create a reference transcriptome assembly using Trinity v2.0.5 with default settings (*Grabherr et al., 2011*). To reduce

redundancy, transcripts were clustered using CD-HIT-Est (*Fu et al., 2012*) at a cut-off of 95% identity. To remove contamination, we used NCBI's VecScreen. This tool searches for segments that match sequences in the UniVec database. We used BUSCO (*Simao et al., 2015*) to assess the completeness of the transcriptome. This tool searches the transcriptome for a reference set of single-copy orthologs. The metazoan set (odb9) was used as a reference set. All genes in the transcriptome were annotated for gene ontology (GO) using the Trinotate part of the Trinity package. Trinotate uses blastp and blastx to compare the predicted peptide sequences and the reference transcriptome to the uniprot_sprot and the uniprot_uniref peptide databases, performs an hmm-scan on the Pfam-A database and assigns GO terms.

## Differential expression

To identify genes that were overexpressed in the shell-forming mantle tissue, the Trinity RNA-seq pipeline (*Grabherr et al., 2011*) was used. First, the reads for both organs in all four individuals were mapped to the reference transcriptome. Second, the mapped reads were counted to visualize the expression of these transcripts, using the estimation method eXpress (*Roberts & Pachter, 2013*). A high count of mapped reads indicates a high expression rate, while a low count of mapped reads indicates a low expression rate. With the estimated counts of reads, expression profiles were generated using the R-package EdgeR (*Robinson, McCarthy & Smyth, 2010*). Normalization took place as part of the EdgeR workflow. The profiles were then filtered on fold change and false discovery rate (FDR) with the Trinity default cut-off scores of 4 and 0.001 respectively. The genes that were overexpressed in the mantle were selected for further analysis. A heatmap of differentially expressed genes versus samples was produced using the script analyze_diff_expr.pl within the Trinity package.

## Sequence variants

To identify consistent sequence differences between the two phenotypes (yellow/banded and the brown/unbanded), we conducted variant calling following the following protocol. First, reads were mapped to the reference transcriptome using Bowtie2 (*Langmead & Salzberg, 2012*). Next, GATK's HaplotypeCaller (*McKenna et al., 2010*) was run to search for single-nucleotide polymorphisms (SNPs). The default parameters were used with a stand_call confidence of 20.0 and a stand_emit confidence of 20.0. The VariantFiltration tool was used with default parameters to filter out false-positive variants. Clusters were generated when 3 SNPs were present within a window of 35 bases. Variants with a Fisher-strand Score (FS) greater than 30.0 or a quality by depth (QD) value less than 2.0 were filtered out. SNPs with sequencing depth less than 10 in any of the samples were removed. The effect of each SNP on the transcript product was predicted using SnpEff (*Cingolani et al., 2012*).

The resulting set of SNPs was filtered for consistency with the phenotypes using GATK and custom R-scripts. The inheritance of the polymorphism is well understood (*Cain & Sheppard, 1954*; *Murray, 1963*; *Jones, Leith & Rawlings, 1977*). The brown ground colour is dominant over the yellow ground colour and absence of banding is dominant over the presence of multiple bands. The candidate SNPs were therefore filtered so that the brown,

**Table 1** Overview of number of reads and GC content per sample and tissue.

| Sample | Individual | Tissue | Number of reads | Number of bases | GC Content |
|--------|-----------|--------|-----------------|-----------------|------------|
| 11 | 1 | Foot | 71,694,188 | 6,273,241,450 | 44% |
| 12 | 1 | Mantle | 71,670,338 | 6,271,154,575 | 46% |
| 13 | 2 | Foot | 77,053,092 | 6,742,145,550 | 43% |
| 14 | 2 | Mantle | 66,729,632 | 5,838,842,800 | 46% |
| 15 | 3 | Foot | 63,903,422 | 5,591,549,425 | 46% |
| 16 | 3 | Mantle | 69,572,264 | 6,087,573,100 | 49% |
| 17 | 4 | Foot | 121,731,962 | 10,651,546,675 | 46% |
| 18 | 4 | Mantle | 139,455,234 | 12,202,332,975 | 49% |

unbanded snails were either heterozygous or homozygous and the yellow, banded snails were homozygous. Mantle-enriched transcripts contained at least one phenotype-consistent sequence variant were blastn-searched (*Altschul et al., 1990*) against the non-redundant nucleotide database.

## RAD-tags

*Richards et al. (2013)* identified eleven anonymous RAD tag sequences that lie near the colour-polymorphism supergene. In the original research, a cut-off of 96 base pairs was used for the tags. Because of this, the tags are shorter than the full assembly of the RAD tag reads. The overlapping parts of the RAD tag reads were used to generate a consensus for each tag that was longer than the original consensus RAD tag. These extended tags were blast-searched against the reference transcriptome using the megablast algorithm within the Blast+ (*Camacho et al., 2009*) command line tool. Hits were filtered by $E$-value less than $10^{-10}$.

## RESULTS

### Transcriptome assembly and annotation

Eight RNAseq libraries (two for each individual) were constructed and sequenced, yielding between $63.9 \times 10^6$ and $139.5 \times 10^6$ paired reads per sample (Table 1). These reads were assembled into contigs representing 171,051 putative transcripts deriving from 33,109 putative genes (Table 2). The frequency of the number of transcripts per gene is shown in Fig. S2. Clustering using CD-HIT reduced the number of transcripts to 150,380. On average, 71% of the raw sequence reads aligned at least once to the reference transcriptome (Table S1). After removing vector contamination, 147,397 transcripts remained.

The completeness of the transcriptome was assessed using BUSCO. Of the 978 single-copy orthologs that were searched for, 920 were found completely. Of these, 765 (78.2%) were found single-copy, 155 (15.8%) orthologs were found duplicated. A further 37 (3.8%) of the orthologs were found fragmented and 21 (2.2%) were not found in the transcriptome.

A total of 111,416 (75.6%) transcripts were annotated by Trinotate. Protein sequence comparison to the uniprot_sprot database yielded annotations for 89,386(60.6%) transcripts, to the uniprot_uniref database for 87,045 (59.1%) transcripts and to the Pfam

**Table 2** Statistics of the transcriptome assembly.

| Statistic | Number | Number after filtering |
|---|---|---|
| Number of contigs | 171,051 | 147,411 |
| Average contig length | 847.86 bp | 783.47 bp |
| Median contig length | 537 bp | 515 bp |
| Number of genes | 33,109 | 25,334 |
| N50 | 1,111 | 968 |
| GC Content | 42.20% | 42% |
| Total bases | 145,027,740 | 115,481,543 |

**Table 3** Numbers of SNPs that were homozygous in the yellow snails and the number of transcripts these were found in. These are further broken down into sets that showed allelic patterns that were consistent with the shell phenotypes of the sampled snails, differentially expressed in mantle tissue or both.

| Property | Number of SNPS | Number of transcripts |
|---|---|---|
| Total number | 73,817 | 17,499 |
| Consistent | 5,776 | 2,592 |
| Differentially expressed | 4,992 | 817 |
| Differentially expressed and consistent | 569 | 197 |

database for 30,405 (20.6%) transcripts. After annotation, Trinotate assigned GO-terms to 19,658 genes, covering 26,039 transcripts (17.7%; Data S1).

## Differential expression

Hierarchical clustering of the overall expression data clearly separated the mantle and foot tissue samples (Fig. 1). EdgeR identified 1,961 transcripts as upregulated and 1,260 as downregulated in the mantle relative to the foot (Fig. 2, Data S2).

## Sequence polymorphisms

A total of 73,817 SNPs passed our filtering steps (Table S2). This included 12,273 synonymous and 12,451non-synonymous variants. The remaining 49,092 SNPs fell outside predicted open reading frames. A total of 569 SNPs were found in transcripts that were overexpressed in the mantle and showed phenotype-consistent allelic variation (Table 3). A total of 197 mantle-enriched transcripts contained at least one phenotype-consistent sequence variant. This set was subjected to more detailed annotation. We obtained database matches for 98 transcripts, of which 38 were functionally annotated (Table S3). Annotations that indicate putative functions related to shell formation are summarized in Table 4. Prominent among these annotations are sequences that are thought to play a role in the production of the nacreous layer in mollusc shells, including dermatopin and nacroperlin genes (marked in Table 4).

## RAD-tags

A set of 12 transcripts had a match with at least one of the elongated tags that lie in close proximity to the supergene (*Richards et al., 2013*) (Table S4). None of these transcripts were found in our list of differentially expressed transcripts with phenotype-consistent

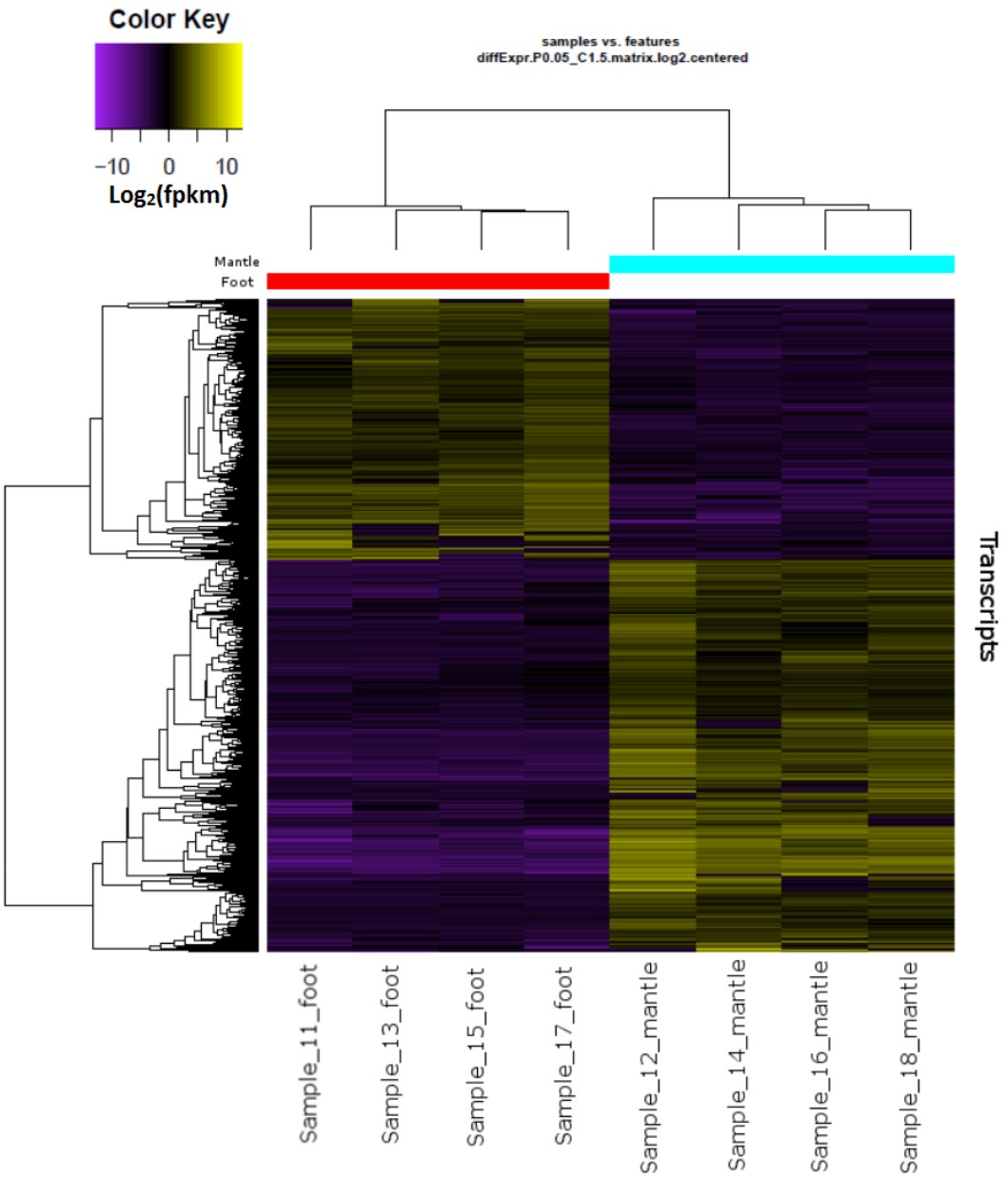

**Figure 1   Overall gene expression differences according to tissue and individual snail.**

variants. Eight of the twelve transcripts (matching two of the RAD tags closest to the supergene) had a blast hit to the nr and nt databases (Table S4, Data S3). All of these hits were the *Helix pomatia* homolog of metallothionein. This same sequence was also found in the variant analysis (six of 197 transcripts = 3%). None of the transcripts in these two sets overlap. The percentage of metallothionein hits among the entire reference transcriptome was 6,1% (2,506 of 40,748 transcripts), suggesting that this gene is not overrepresented in our variant analysis.
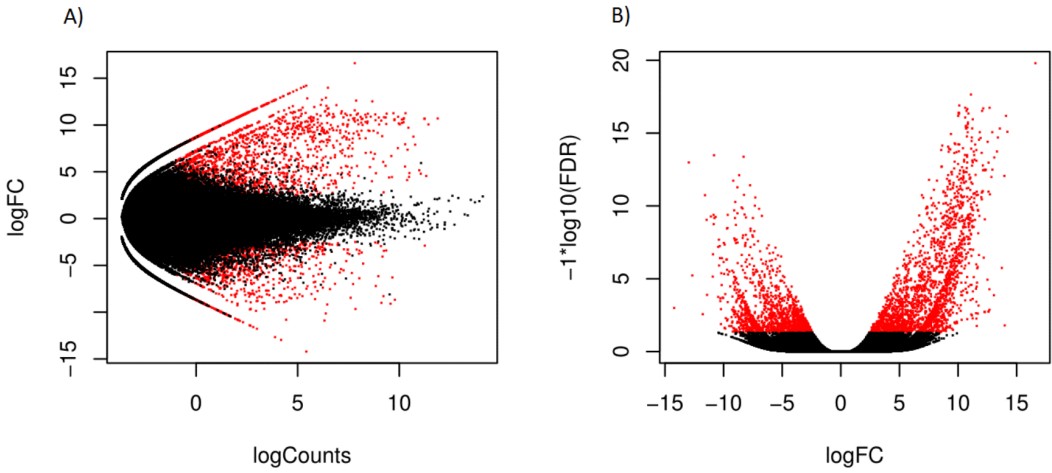

**Figure 2** Scatterplot (A) log counts versus log fold change and volcano plot (B) log fold change versus statistical significance, for differential expression in mantle tissue versus the foot tissue, obtained using EdgeR. Transcripts marked in red were considered differentially expressed.

**Table 4** The most informative annotations from Table S3. Transcripts with putative function in nacre and shell production are marked in italics.

| Contig name | Functional annotation | Effect |
|---|---|---|
| c280576_g1_i1 | *Biomphalaria glabrata* glycine and methionine-rich –like | synonymous_variant |
| c280925_g1_i2 | *Parasteatoda tepidariorum* keratin-associated 6-2-like | synonymous_variant |
| c350256_g1_i1 | *Anas platyrhynchos* BPI fold-containing family B member 3 | synonymous_variant |
| c368572_g1_i1 | *Aplysia californica* epithelial splicing regulatory 1-like | synonymous_variant |
| c369765_g4_i2 | *Biomphalaria glabrata* ferric-chelate reductase 1-like | synonymous_variant |
| c371799_g2_i1 | *Biomphalaria glabrata* sushi, von Willebrand factor type A, EGF and pentraxin domain-containing 1-like | synonymous_variant |
| *c264073_g1_i1* | *Camelus bactrianus mucin-2-like* | *frameshift_variant & stop_gained inframe_insertion* |
| *c366293_g1_i1* | Biomphalaria glabrata mucin-2-like | *missense_variant* |
| *c321814_g1_i1* | *Euhadra herklotsi mRNA for Dermatopontin1* | *intergenic_region* |
| *c355427_g1_i1* | *Euhadra herklotsi mRNA for Dermatopontin1* | *stop_gained missense_variant* |

## DISCUSSION

Our analysis identified a list of 300 candidate transcripts that were differentially expressed in the shell-forming mantle tissue and contained SNP patterns that matched the shell phenotypes. For most of these transcripts, it was impossible to infer their role in shell formation as only ~20% of these genes were functionally annotated. Furthermore, the supergene may consist mostly of regulatory genes without a previously identified role in shell biosynthesis. However, two sets of transcripts could be putatively linked to shell or pigment production.

The first set consists of transcripts involved in the synthesis of the nacreous layer in the snail's shell. Mollusc shells consists of three layers: the outer prismatic layer, the

inner prismatic layer and the nacreous layer (*Suzuki & Nagasawa, 2013*). It is thought that dermatopontin, a major shell matrix protein, is involved in the production of the nacreous layer (*Jiao et al., 2012*). We found two hits to the *Euhadra herklotsi* ortholog of this gene. Two other transcripts showed resemblance to Mucin 2, which is a gene that has a mollusc homolog that is thought to be involved with production of the nacreous layer in molluscs (*Marin et al., 2000*). The mollusc variant of this gene is nacroperlin, which is found in shell of Mediterranean mussels (*Marin et al., 2000*). The nacreous layer is the innermost layer of the snail's shell and is therefore unlikely to directly affect pigmentation, however differences in the nacreous layer may affect other shell traits that differ between the morphs. Shell strength is known to differ between *C. nemoralis* colour morphs, with pink shells stronger than yellow shells and banding stronger than no banding (*Jiao et al., 2012*; *Rosin et al., 2013*).

The second set of transcripts consists of transcripts that produce metallothionein. Metallothionein is a lightweight thiol-rich protein, the production of which is induced by the presence of heavy metals such as zinc, copper or cadmium. This molecule inhibits the production of melanin following oxidative stress (*Sasaki et al., 2004*). Melanin is a well-known pigment that is found in many tissue types and often has a black or brown colour. An inhibition of the production of melanin can possibly lead to a lack of pigmentation in the tissue. This might contribute to the banding pattern or the ground colour of the shell. Melanin pigments are produced in organelles called melanocytes, which contain a subcellular zinc reservoir. This zinc can trigger a reaction with metallothionein to reduce the production of melanin (*Borovanský, 1994*). The density of melanocytes in the snail mantle was found to be correlated with darkness of the lip and possibly the banding pattern on the shells (*Emberton, 1963*). The ground colour of the shell was not correlated with the density of melanocytes. This suggests that the metallothionein transcripts we identified could be involved in the production of the banding pattern and that they are less likely to be involved in the ground colour.

The aim of this research was to find candidate genes underpinning the *Cepaea* shell colour polymorphism. Due to the modest sample size, our power to detect differential expression was limited. Furthermore, we found a relatively high ratio of synonymous to nonsynonymous SNPs (close to 1:1), which was probably the result of over-prediction of coding regions in partial transcripts and transcripts overlapping introns. This can result in a higher number of predicted non-synonymous SNPS (*Lopez-Maestre et al., 2016*). Nevertheless, we identified 300 candidates that showed mantle-specific expression and phenotype-consistent SNP patterns. In addition to these, we found fifteen transcripts matching RAD-tag sequences that are associated with the shell-colour supergene. Functional annotation of these transcripts should be an excellent starting point for elucidating the molecular underpinning on the *Cepaea* colour polymorphism.

## ACKNOWLEDGEMENTS

We thank Heike Kappes for performing the snail dissections, Emile de Meijer and Henk Buermans for RNA extraction and library preparation, and Peter Neleman, Mirna Baak and Patrick Wijntjes for their earlier analysis of this dataset.

### Funding

This project was funded by a grant from the Fonds Economische Structuurversterking awarded to Naturalis Biodiversity Center. The funders had no role in study design, data collection and analysis, decision to publish, or preparation of the manuscript.

### Grant Disclosures

The following grant information was disclosed by the authors:
Naturalis Biodiversity Center.

### Competing Interests

The authors declare there are no competing interests.

### Author Contributions

- Jesse Kerkvliet performed the experiments, analyzed the data, contributed reagents/materials/analysis tools, wrote the paper, prepared figures and/or tables, reviewed drafts of the paper.
- Tjalf de Boer performed the experiments, analyzed the data, contributed reagents/materials/analysis tools, reviewed drafts of the paper.
- Menno Schilthuizen conceived and designed the experiments, reviewed drafts of the paper.
- Ken Kraaijeveld conceived and designed the experiments, contributed reagents/materials/analysis tools, wrote the paper, prepared figures and/or tables, reviewed drafts of the paper.

### Data Availability

Reads: SRA, SRP101411.

Transcriptome assembly:

TSA: GFLU00000000.

BioProject: PRJNA377398.

### Supplemental Information

Supplemental information for this article can be found online at http://dx.doi.org/10.7717/peerj.3715#supplemental-information.

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
