# Peer review of "Candidate genes for shell colour polymorphism in Cepaea nemoralis"

_PeerJ, doi:10.7717/peerj.3715_

## Round 0.1 · original submission · Major Revisions

Concerns raised by reviewer#1 and 2 on your manuscript about reason for finding high non-synonymous SNPs your study should be adequately discussed. In addition, reviewer#3 concern on confidence for genes reported as differentially expressed should be addressed as suggested.

·

Basic reporting

no comment

Experimental design

no comment

Validity of the findings

no comment

Additional comments

The authors provide a well-designed study that contributes essentially to the understanding of the genetic mechanism underlying the colour polymorphism in Cepaea. Anything is clearly explained and I could not find any obvious flaws.
The only issue that is curious is that the authors reported 9,330 synonymous and 27,166 non-synonymous SNPs (line 176-177). Is this correct? I would expect that there a much more synonymous than non-synonymous variants as a result of purifying selection. Perhaps the authors can discuss reasons for the high surplus of non-synonymous SNPs.

The experimental design of the study was straightforward. The identification of genes that were differentially expressed in the shell-forming mantle tissue and the comparison of the genotypes with the phenotypes under consideration of the inheritance of the characters is an appropriate approach to identify candidate genes that may be involved in the genetic mechanism resulting in the observed phenotypes. The searching for matches between the sequenced transcripts and the elongated tags that lie in close proximity to the supergene
is also appropriate. Unfortunately, none of the transcripts that match with at least one of the elongated tags that lie in close proximity to the supergene was found in the list of differentially expressed transcripts with phenotype-consistent variants. However, this does not mean that the findings are not sound. Obviously, more research is necessary. Perhaps, the authors may state at the end of the manuscript which are the next steps to identify the genes involved in pattern formation that they would propose.

Minor remarks
Line 56. Replace “Schwander et al., 2014” by “Schwander, Libbrecht & Keller, 2014”.
Line 59. Replace “(Mann & Jackson, 2014)” by “Mann & Jackson (2014)”.

Reviewer 2 ·

Basic reporting

no comment

Experimental design

no comment

Validity of the findings

no comment

Additional comments

Kerkvliet and colleagues use transcriptomic methods to identify candidate genes for the colour polymorphism in Cepaea nemoralis. In the first part, the authors identify genes that are up or down regulated in the mantle relative to the foot tissue, theorising that upregulated genes are possible candidates. In the second part, the authors identify which of these upregulated genes show polymorphism that is in phase with the colour polymorphism, thus refining the initial list down to 197 different candidates. Some of the genes are identifiable, based on previous information, but the majority are not. Of the genes that have known function, a substantial number may be involved in shell biosynthesis. Finally, the authors compared the transcriptome to a set of previously identified anonymous RAD-tags – 12 transcripts had a blast hit to the RAD-tags, though none of these transcripts were identified as differentially expressed or showing in phase polymorphism.

* more important points

L79 “as it is possible that these tag sequences are themselves part of the supergene”. This is not likely since Richards et al report that no marker is in perfect linkage with the colour/banding part of the supergene (though I suppose it is formally possible that the RAD locus is in perfect linkage with other loci in the supergene)

L87 Perhaps it is the colour reproduction, the two darker shells appear to be pink, not brown.

*L188 onwards. A lack of power is a significant concern in the differential expression section – and I think that the authors should acknowledge this. More specifically, there are some further straightforward analyses that the authors could do to investigate these data.

What are the expression patterns of the transcripts that show differential expression? Given the small sample size, it is likely that many of the transcripts that show significantly different expression levels above the FDR are presence/absence - which is perhaps less convincing. How many of the statistically significant differences are presence/absence?

Would it be possible to randomise the data and see how many significant differentially expressed transcripts you find? If this showed very few significant hits, then the main finding would be more convincing.

*L193 Important point – there are 3 times as many non-synonymous SNPs as synonymous SNPs! If this is real, then what is the explanation? If it is an artifact, then how has it come about? I am unable to think of any reasonable explanation as to why you might get a whole genome excess of non-synonymous SNPs, and so I find this quite troubling. Surely, this is a typo – it should read the reverse? "This included 9,330 NON-synonymous and 27,166 synonymous variants".

L194 “The remaining 37,322 SNPs fell outside predicted open reading frames” – it is not clear to me if these SNPs were used further or not? For example, is it 5776 consistent SNPs out of 73817 or 5776 SNPs out of (73817-37322)?

*L200 By identifying genes with putative functions in shell biosynthesis, the authors argue that they have found candidate genes for the shell polymorphism. I think that this argument needs some further elaboration.

In the first part, differential expression is used to find genes that are associated with shell biosynthesis, in all of its forms. In the second part, this subset of genes is inspected to find those that may be associated with polymorphism – the set of genes that are both are thus candidate genes for the polymorphism.

I don’t think that there is any reason to suppose that the genes in the supergene will necessarily be associated with known shell biosynthesis functions – they might just be regulatory genes (although of course, it is inevitable that we focus on those of known function). Perhaps this caveat needs to be included in the text?

*L207 Further information is required for these matches, ideally alignments in the supplementary information? As presented, the data are not convincing – there are likely some repetitive regions in the RAD loci, and so my concern is that these transcripts are hitting those repeats.

Annotated reviews are not available for download in order to protect the identity of reviewers who chose to remain anonymous.

Reviewer 3 ·

Basic reporting

Mostly clear

Experimental design

Original. Experimental design could be further developed

Validity of the findings

No comment

Additional comments

This research paper did investigate the tissue transcriptomes of the body and mantle of Cepaea nemoralis with 2 different shell phenotypes. Comparison of the mantle with body, and the snails of different phenotype have provided a list of candidate genes involved in shell color based on differential expression and similarity with other known genes involved in biomineralization. This investigation also provides RAD tag outcomes regarding candidate genes that may be associated with the shell supergene, although the numbers provided are confusing.

The most significant concern with the research paper relates to the confidence for those genes reported as differentially expressed. This could be much improved through increased sample size for RNAseq and/or additional expression experimentation such as qPCR or protein MS analysis. It would also have been of greater significance to include expression analysis for the other shell banding phenotypes. Other comments I have that should be fixed or considered include:

Abstract
L20 – I’m not sure that the ‘this polymorphism’ has been indicated in the prior text.
L25 – Its not clear how ‘Combining the results’ of what had been described could yield the 197 candidate transcripts.
L28 – define ‘RAD’
L28 – Says 7 transcripts, yet Table S4 lists 10 that annotate as metallothionein. Then, in the results it indicates 8. Why the variation?


Introduction

L62 and L64 – is it shell polymorphism?
L78 – The 4 juveniles don’t all have different phenotypes, but rather there are four juveniles representing two different phenotypes
L81 – what really defines the ‘body’? I assumed that it was everything except the mantle, yet there were remains that were stored in alcohol.
L84 - What kit was used for polyA isolation and cDNA synthesis, or is there a reference since standard protocols can be very different for this.
L85 – what are paired-end truseq libraries? Usually sequencing is paired-end, not the library.
L87 – where was the HiSeq sequencing undertaken?

Results

L156 – there is no fig. S3 but there is a fig. S2 not indicated.
Table 1: what’s the significance of the sample numbers 11-18? I don’t see that this is clearly explained in the text.
The details of gene GO results should be provided in the supplementary.
Figure 1 – what does the scale bar represent?
Figure 2 – what is MA?
The differentially expressed genes should be provided in the supplementary.
Fig S1 – again, what is the meaning of the numbers shown?
Table S2 – I don’t see how the numbers specified in the text correspond to those in this table. Eg. In main text it says 73,817 SNP total, while in the Table it has 461,799 SNP total.
Table 4 – provide references for transcripts implicated with putative function in nacre and shell production.
Table S4 – has 15 contigs listed, yet the main text indicates 12

L211 – need a reference for: Two other transcripts showed resemblance to Mucin 2, which is another gene that is thought to be involved with production of the nacreous layer in molluscs.

L216 – I’m unclear of the relevance of the sentence starting: Shell strength is known…..
L221 – is metallothionein a protein or a molecule?

---

## Round 0.2 · Minor Revisions

Please discuss about the issue raised by one of the reviewer on numbers of synonymous and non-synonymous SNPs.

Reviewer 2 ·

Basic reporting

I have no further comments

Experimental design

I have no further comments

Validity of the findings

I have no further comments

Additional comments

I understand that there was an error in the original manuscript that has now been corrected. However, I am still concerned that equal numbers of synonymous and non-synonymous SNPs were recovered.

Is this a) an artifact of the method or b) biological interesting e.g. due to selection? Worthy of discussion at the very least.

---

## Round 0.3 · accepted · Accept

The revised manuscript has addressed all remaining issues. Now it is ready for publication.